# Pairwise Causality Guided Transformers
# for Event Sequences

**Xiao Shou**
RPI
xshou01@gmail.com

**Debarun Bhattacharjya**
IBM Research
debarunb@us.ibm.com

**Tian Gao**
IBM Research
tgao@us.ibm.com

**Dharmashankar Subramanian**
IBM Research
dharmash@us.ibm.com

**Oktie Hassanzadeh**
IBM Research
hassanzadeh@us.ibm.com

**Kristin Bennett**
RPI
bennek@rpi.edu

## Abstract

Although pairwise causal relations have been extensively studied in observational longitudinal analyses across many disciplines, incorporating knowledge of causal pairs into deep learning models for temporal event sequences remains largely unexplored. In this paper, we propose a novel approach for enhancing the performance of transformer-based models in multivariate event sequences by injecting pairwise qualitative causal knowledge such as 'event Z amplifies future occurrences of event Y'. We establish a new framework for causal inference in temporal event sequences using a transformer architecture, providing a theoretical justification for our approach, and show how to obtain unbiased estimates of the proposed measure. Experimental results demonstrate that our approach outperforms several state-of-the-art models in terms of prediction accuracy by effectively leveraging knowledge about causal pairs. We also consider a unique application where we extract knowledge around sequences of societal events by generating them from a large language model, and demonstrate how a causal knowledge graph can help with event prediction in such sequences. Overall, our framework offers a practical means of improving the performance of transformer-based models in multivariate event sequences by explicitly exploiting pairwise causal information.

## 1   Introduction

Multivariate event sequences are datasets where different types of events occur sequentially, without meaningful time stamps. They can be viewed as categorical time series datasets where exactly one event from a known set of event types occurs at any position in a sequence. Applications of such datasets abound in various fields, including the social sciences, healthcare, finance, advertising, and engineering. Mining patterns from event sequences often provides meaningful insights for prediction, detection and analysis [1]; for example, analysis of such sequences is performed to study patterns of medication use or disease progression in healthcare, and to understand stock price prediction and fraud detection in finance. Modeling events is of interest in numerous fields besides data mining; for instance, there has been a surge of interest over the last several years around modeling time-stamped event streams in machine learning, e.g. [2, 3, 4], as well as in leveraging events for various tasks in natural language processing, e.g. [5, 6].

The field of machine learning has broadly experienced substantial attention around augmenting existing data with various forms of background knowledge, an area that some have referred to as *informed machine learning* [7]. A body of prior work has shown how incorporating various forms of domain knowledge could potentially aid deep learning models in performing better, more efficiently,

37th Conference on Neural Information Processing Systems (NeurIPS 2023).

and more reliably [8], especially when the amount of available data is low. Such knowledge could also help make the models more interpretable and easier to understand, which is often important in critical real-world applications such as healthcare [9]. A commonly explored direction is one where physical rules are injected to control model prediction in physics regulated systems [10]. Knowledge augmentation is also being pursued in the setting of pretrained large language models [11]. There have been some recent related forays for machine learning of event models, such as through leveraging temporal logical rules [12] and data-dependent knowledge measures [13]. A recent study considers injecting qualitative structural statements in score-based learning systems for temporal point processes, which results in closed form and interpretable score-based loss [14].

In this paper, we address an important setting that brings together aspects of knowledge injection with modeling event sequences: how to effectively augment domain knowledge while learning multivariate event sequences using neural models. Specifically, we consider background knowledge of the form of qualitative statements such as: 'event type Z amplifies (or inhibits) event type Y'. Such statements embed the notion of pairwise causality and can be better paraphrased as 'occurrences of event type Z make future occurrences of event type Y more (or less) likely'. Such causal knowledge can oftentimes be acquired directly from causal knowledge graphs [15, 16, 17, 18] or learned from data.

Integrating pairwise causal statements into learning neural models for temporal event sequences poses at least two major technical challenges that need to be suitably tackled:

1. Causal inference usually entails examining both the *factual outcome* as well as the *counterfactual outcome* for a binary treatment. Since we consider a specific type of non-i.i.d. temporal setting where events occur sequentially, one cannot directly apply the standard potential outcome framework [19] without appropriate definitions that account for the sequential nature of the data.

2. Pairwise causal statements about event types reflect a type-to-type global property that may or may not have been inferred from instance-level data. It is relatively straightforward to use such information through additional regularization terms for loss functions to learn typical parametric models; this is because the (learned) parameters of such models are global and stationary, and thus any deviations from the statement can be penalized. However, a neural event sequence model such as an autoregressive transformer captures the dynamics of event instances, thus requiring a different approach for injecting type-level knowledge.

**Contributions.** In this paper, we incorporate causal knowledge augmentation into the modeling of temporal event sequences. Specifically, we propose pairwise causality guided learning in transformers that suitably addresses the afore-mentioned challenges. Our contributions are as follows: 1) We formulate the problem of incorporating pairwise causal knowledge into neural autoregressive models in the context of temporal event sequences, proposing suitable definitions for a formal causal inference investigation. 2) We provide theoretical justification for our procedure, particularly to combat the issue of time confounding and to achieve representational balance. 3) We conduct a detailed empirical evaluation demonstrating superior predictive performance as compared to state-of-the-art baselines on synthetic datasets as well as real-world benchmarks. 4) We explore a novel application involving large language model generated event sequences, illustrating both how a causal knowledge graph could aid in event prediction, as well as how one could potentially build event models through knowledge injection without relying on existing event sequences.

## 2 Related Work

Our work is interdisciplinary in nature, touching upon topics such as event sequence modeling, causal inference, neural networks, and natural language processing. We briefly review closely related work across some of these topics.

**Markov Models.** Markov models have been widely deployed over several decades for analyzing temporal event sequences. Besides the classic $k$th-order Markov models where the probability of the next state depends solely on the previous $k$ positions in the sequence, other variations include (but are not limited to) hidden Markov models [20], models incorporating linear combinations [21], variable order Markov models [22], summary Markov models [23], etc. While such models have advantages around interpretability and low data requirements, they are often unable to capture long-range and potentially complex historical dependencies, unlike neural sequence models.

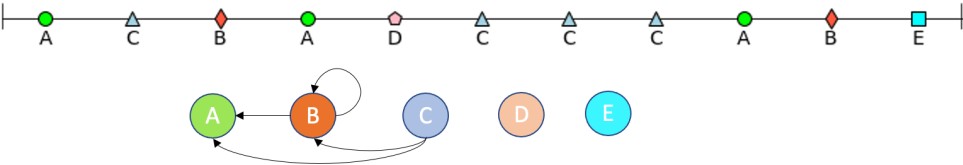

Figure 1: An illustrative event sequence over labels $\{A, B, C, D, E\}$ where event pair A and B are of interest. An example of event labels from healthcare is as follows: $A$: Emergency visit, $B$: Cardiac event, $C$: Medication refill, $D$: Allergy diagnosis, $E$: Anxiety diagnosis.

**Neural Sequence Models.** RNNs are a class of neural networks that process sequences of inputs. In an RNN, the output at each time step depends not only on the current input but also on the previous output. Variants of RNNs such as Long Short-Term Memory (LSTM) and Gated Recurrent Units (GRU) have been shown to be effective in various applications, including natural language processing, speech recognition, and time-series analysis. State-of-the-art sequence models involve the use of attention-based transformers. Current research focuses on making more reliable predictions from factual associations, for example in large language models [24], and on equipping such transformer-based models with arithmetic, commonsense or symbolic reasoning [25]. The importance of causal reasoning for natural language understanding has also recently emerged [26].

**Event Models.** Event sequences are a popular source of data for numerous data mining algorithms [1, 27, 28]. There is also a long line of research around modeling time-stamped streams using temporal point processes [29] that has become popular in machine learning. Graphical event models (GEMs), also known as local independence graphs, are a family of graphical models that represent temporal point processes [30, 2, 3]. GEMs capture the conditional independence structure of the continuous-time dynamics of event processes and are closely related to the notion of Granger-causality [31, 32, 33]. A recent Bayesian approach uses a score-based framework for injecting qualitative statements while learning a sub-family of parametric GEMs [14]. Our work here differs in that we model event sequences without time-stamp information, and inject pairwise causal knowledge into transformer models, which are known to be more flexible and therefore better than typical parametric models at tasks such as event prediction [34].

**Sequential Treatment Effect Estimation.** Substantial research has been conducted on time-varying treatment estimation for longitudinal observation data [35, 36], and many of the state-of-the-art approaches leverage neural models [37]. Some prominent examples over the last couple of years include recurrent marginal structural networks [38], counterfactual recurrent networks [39], G-Nets [40] and causal transformers [41]. In addition, some recent work has studied causal inference for temporal process data [42, 43], by leveraging time information from event stream datasets. This line of work does not consider injecting pairwise causal knowledge into neural temporal event sequence models.

**Causality in NLP.** Feder et al. [44] provide an excellent survey of work done at the intersection of causal inference and NLP. There is also a body of work on the extraction of causal relations from natural language descriptions [45]. Causal knowledge derived from text either through inference or extraction can be used for the injection of pairwise causal knowledge into event models.

## 3 Causal Event Pairs in Event Sequences

In this section, we first introduce basic notation and terminology and then motivate how one can incorporate pairwise causal knowledge in neural autoregressive models such as attention based transformers for temporal event sequences.

### 3.1 Event Sequences & History-Confounding Dynamics

Consider a multivariate **event sequence dataset** $\mathbf{D}$ which consists of $K$ sequences: indexed by $k$, each sequence takes the form of $\mathbf{D_k} = [l_i]_{i=1}^{N_k}$, where event label (or type) $l_i$ at position $i$ in the

sequence is from a discrete label set, $l_i \in \mathcal{L}$. The total number of events and event types are denoted $N$ and $M$ respectively, i.e. $N = \sum_{k=1}^{K} N_k$ and $M = |\mathcal{L}|$. Let $h_i = \{l_j\}_{j=1}^{i}$ denote the history of past events up to $i$. Consider the following example in a typical event sequence:

**Example 1** *Figure 1 shows an example event sequence with 5 event labels, $\mathcal{L} = \{A, B, C, D, E\}$. Consider the 3 causal pairs $(A, B), (B, C), (A, C)$ potentially provided by a domain expert. B causes the probability of future occurrences of A to increase, denoted as $B \nearrow A$. Another 'competing' event C causes the probability of future occurrences of A to decrease, i.e. $C \searrow A$. Event C also excites B, i.e. $C \nearrow B$. Note that it can be challenging to determine the influence of how one event affects another in an observational temporal event sequence.*

Example 1 entails pairwise qualitative statements for sequential events occurring over time, and therefore requires us to establish some formal tools; we turn to causal inference and the potential outcome framework [19] in particular to address this issue. We formulate treatment, covariate and outcome models for any event sequence $\mathbf{D}_k$ as follows. Note that we abuse notation slightly and sometimes use lower case (e.g. $y$, $z$) for event labels to indicate that they can also be viewed as instances of random variables.

**Definition 1** *Formulation of Causal Event Pair: For a pair of event labels $(z, y)$, the binary **treatment** variable $Z_i$ at time $i$ is whether or not label $z$ occurs at position i, i.e., $l_i = z$ or $l_i \neq z$. For a specific $Z_i$ at time $i$, the corresponding **covariates** are historical occurrences of event labels including z , i.e. $h_{i-1}$. The **outcome** is the probability of occurrence of $y$ (at least once) in the next window $w$ given the treatment and covariates, i.e., $p(I(y)_i^w|l_i, h_{i-1}) = 1 - p(l_{i+1} \neq y, l_{i+2} \neq y, ..., l_{i+w} \neq y|l_i, h_{i-1}) = 1 - p(l_{i+1} \neq y, l_{i+2} \neq y, ..., l_{i+w} \neq y|h_i)$ for $w \in \{1, 2, 3, ..., W\}$ where $W \ll N_k$.*

In the above definition, $I(y)_i^w$ is an indicator function for occurrence of $y$ (at least once) within the next window $w$ from position $i$. Many other choices of outcome models are possible. For instance, one could choose the next arrival time of $Y$, however we may not observe $Y$ at all in a finite time horizon. The benefit of our definition is that one can easily compute the probability of $Y$ in the next window $w$. The simplest setting is when the window $w = 1$, which we will use as default. For cases where $w > 1$, the same principle holds and we discuss this variation in Section 4.3. We define the propensity score as follows.

**Definition 2** *Under Definition 1, a valid propensity score is defined as $p(l_{i+1} = z|h_i)$.*

It is challenging to directly compute the propensity score under the raw history vector $h_i$ due to the varying length of history. Furthermore, such history can be of high dimension for large $N_k$. A possible approach is to compute the propensity score in its latent representation. A neural autoregressive model with parameters $\theta$ is trained to maximizes its log likelihood and we thus obtain its propensity score $p_{\theta*}(l_i = z|h_{i-1})$ at each time instance $i$ with a sample/batch size $B$:

$$\theta^* = argmax_\theta \mathbb{E}_{B \sim p(D_k)}[\sum_{j=1}^{|B|} \sum_{i=1}^{N_k-1} p_\theta(l_{i+1}|h_i)] \tag{1}$$

**Transformer for Event Sequences.** Consider an autoregressive transformer $G$ which takes an event (sub)sequence and outputs a sequence of categorical probability distribution: $\mathcal{H} \to \mathcal{Y}$. An event (sub)sequence $H$ is first embedded into $\mathbf{X}$ via position embedding. The two major components of a self attention layer are attention layers (Attn) and position-wise feed-forward networks (FFN) via residual connection. We show the $h$-head attention with upper triangular attention mask $\mathbb{M}$ so that only future events attend to the past in our setting:

$$Attn(\mathbf{X}) = \mathbf{X} + \sum_{i=1}^{h} \mathbf{W}_O^i \mathbf{W}_V^i \mathbf{X} \cdot (\sigma[(\mathbf{W}_K^i X)^T \mathbf{W}_Q^i \mathbf{X}] \odot \mathbb{M}),$$

$$FFN(X) = Attn(X) + \mathbf{W}_2 \cdot ReLU(\mathbf{W}_1 \cdot Attn(\mathbf{X}) + \mathbf{b}_1 \mathbf{1}^T) + \mathbf{b}_2 \mathbf{1}_n^T, \tag{2}$$

where $\mathbf{W}_Q^i$ $\mathbf{W}_K^i$ $\mathbf{W}_V^i$ and $\mathbf{W}_O^i$ are query, key, value and output matrix in the $i^{th}$ head with appropriate dimension. $\sigma$ is the softmax operation.

**Theorem 2** *Under Definition 1, an autoregressive transform $G$ outputs a valid propensity score, i.e., $p(l_{i+1} = z|h_i)$. A valid propensity score is computed by using only past history.*

Proof Sketch. The only operations that are not point-wise are in the attention module. Let $\tilde{\mathbf{X}} = \mathbf{W}_O^i \mathbf{W}_V^i \mathbf{X}$ and $\tilde{\mathbf{A}} = \sigma[(\mathbf{W}_K^i X)^T (\mathbf{W}_Q^i \mathbf{X})] \odot \mathbb{M}$. Thus for each head $i$, $(\tilde{\mathbf{X}}\tilde{\mathbf{A}})_{ij} = \sum_k \tilde{\mathbf{X}}_{ik} \tilde{\mathbf{A}}_{kj}$. Notice masked attention $\tilde{\mathbf{A}}$ is upper triangular, $\sum_k \tilde{\mathbf{X}}_{ik} \tilde{\mathbf{A}}_{kj} = \sum_{k \leq j} \tilde{\mathbf{X}}_{ik} \tilde{\mathbf{A}}_{kj}$ Hence only past history is involved for predicting the next treatment assignment $l_{i+1} = z$.

**Theorem 3** *Under Definition 1, an autoregressive transformer $G$ with $L$ blocks generates a full spectrum of balancing scores at each instance $i = \{1, ..., N_k - 1\}$ for each layer $l = \{1, 2, ..., L\}$.*

Proof Sketch: The proof follows **Proposition** 4 from [46] and Theorem 1 above as Transformer $G$ with $L$ blocks can be viewed as a sequence of compositions, i.e. $G := \sigma(f_L \circ f_L - 1 \circ ... f_1 \circ f_0(H))$, where $f_0$ is the position embedding, $f_l$ is ATTN followed by FFN, and $\sigma$ is the softmax operation.

# 4 Proposed Framework

We propose an approach to inject pairwise causal knowledge, referring to our model as a pairwise-causality transformer for event sequences (PC-TES). More precisely, we assume that such causal pair knowledge holds for the average case, i.e., on average, for every sequence $k$, event type $Z$ decreases/increases the probability of future occurrences of event type $Y$. Formally, this impacts the difference $\mathbb{E}[P(l_{i+1} = y)|l_i = z] - \mathbb{E}[P(l_{i+1} = y|l_i \neq z]$ for arbitrary position $i$ in the sequence. We make the following assumption regarding measured confounding in temporal sequences, which is not uncommon in sequential treatment and precision medicine [39, 41]. This aspect also frequently appears in economic event sequences [23] and diabetic patient activities [47]. While such confounding terms can be trivially handled by identifying other parental states in typical parametric models (see e.g. [14]), it is not straightforward for neural models.

**Assumption 1** *An observational multivariate event sequence under the above framework involves time-varying confounding, i.e., for a causal pair $(Y, Z)$, history not only affects the occurrence of future events $Y$ but also affects the occurrences of $Z$.*

This assumption holds for Example 1 – the effect of $B$ on $A$ is not only confounded by $C$ but also any past history; it also implies that any causal pair $(Z, Y)$ has potential history confounding as long as $Z$ and $Y$ are not process independent [30]. In Example 1, occurrences of $B$ in the past affect both the occurrences of outcome $A$ and $B$. In addition, occurrences of $C$ in history on the other hand affect the occurrences of $A$ and $B$. The effect of $C$ on the occurrences of $A$ are thus confounded by past histories involving $B$ or $C$ itself and needs to be adjusted. A full qualitative statement in our case becomes 'on average, for each sequence $k$, event type $Z$ decreases/increases the probability of future occurrences of event type $Y$ while holding history constant'. The additional three standard assumptions of **consistency**, **sequential ignorability**, and **sequential overlap** can be made similarly to time series observational data [48, 36] to identify the potential counterfactual outcomes in our setting (see Appendix for further details).

## 4.1 Inverse Probability Weighting in Event Sequences

We define 2 relevant terms with respect to the statement '$Z$ reduces the occurrence of $Y$' via inverse probability weighting. Without loss of generality, consider a window $w = 1$.

**Definition 3** *For an event sequence $k$, the instance-level effect (ILE) at position $i$ is the difference between potential probabilities of $Y$ at $i + 1$ for occurrence and nonoccurence of $Z$ at $i$, i.e. $p(l_{i+1} = y|h_{i-1}, l_i = z) - p(l_{i+1} = y|h_{i-1}, l_i \neq z)$.*

**Definition 4** *For an event sequence $k$, the sequence-level effect (SLE) is averaged over all positions of ILEs, i.e. $\frac{1}{n-1} \sum_{i=1}^{n-1} (p(l_{i+1} = y|h_{i-1}, l_i = z) - p(l_{i+1} = y|h_{i-1}, l_i \neq z))$.*

For an observational temporal event sequence, we impose inverse probability weighting to adjust for time confounding: $\hat{\tau} = \frac{1}{n-1} \sum_{i=1}^{n-1} \left( \frac{p(l_{i+1} = y|h_{i-1}, l_i = z)}{p(l_i = z|h_{i-1})} - \frac{p(l_{i+1} = y|h_{i-1}, l_i \neq z)}{p(l_i \neq z|h_{i-1})} \right)$.

**Theorem 4** *Under Definition 1, Assumption 1 and the consistency, sequential ignorability, and sequential overlap assumptions, given a learned transformer model $G$, our inverse probability weighted sequence-level effect (SLE) under $G$ is an unbiased estimator of the true effect $\tau = \mathbb{E}[P(l_{i+1} = y)|l_i = z] - \mathbb{E}[P(l_{i+1} = y)|l_i \neq z]$.*

Proof Sketch: $\hat{\tau}$ can be rewritten as $\frac{1}{n-1}\sum_{i=1}^{n-1}\frac{(l_i=z)p^*(l_{i+1}=y)}{p^*(l_i=z)} - \frac{1}{n-1}\sum_{i=1}^{n-1}\frac{(l_i\neq z)p^*(l_{i+1}=y)}{1-p^*(l_i=z)}$ where $*$ stands for conditioning on the historical representation obtained from $G$, i.e. $H_i$ and $H_{i-1}$ respectively for numerator and denominator. Thus we can show each $\mathbb{E}\frac{(l_i=z)p^*(l_{i+1}=y)}{p^*(l_i=z)} = \mathbb{E}[P(l_{i+1} = y)^{Z_i=1}]$ and $\mathbb{E}\frac{(l_i\neq z)p^*(l_{i+1}=y)}{1-p^*(l_i=z)} = \mathbb{E}[P(l_{i+1} = y)^{Z_i=0}]$ using the law of iterated expectation where the subscripts $Z_i = 0$ and $Z_i = 1$ are for each stratum.

In practice, if the number of event types is large, causing $p^*(l_i = z)$ to be small, we can use standardized stable inverse probability to avoid numerically instability, similar to [42] for temporal event sequences: $\hat{\tau} = \frac{1}{n-1}\sum_{i=1}^{n-1}\left(\frac{p(z)p(l_{i+1}=y|h_{i-1},l_i=z)}{p(l_i=z|h_{i-1})} - \frac{(1-p(z))p(l_{i+1}=y|h_{i-1},l_i\neq z)}{p(l_i\neq z|h_{i-1})}\right)$ where $p(z)$ is a prior belief of probability of $z$ occurring in the sequence $k$.

## 4.2 Incompatibility Framework

Our injected statements are qualitative, and the ground truth effect $\tau$ is usually unknown even for domain experts. Hence we only determine whether it deviates from the statement in practice. To do so, we adopt an incompatibility framework [49, 14] where qualitative knowledge is injected as a loss term. For the statement '$Z$ inhibits $Y$' for a sequence $k$, our proposed combined loss is:

$$L_{tot} = -\sum_{i=1}^{n_k} p^*(l_i) + \alpha\, max\left(\frac{1}{n-1}\sum_{i=1}^{n-1}\left(\frac{\mathbb{1}(l_i = z)p^*(l_{i+1} = y)}{p^*(l_i = z)} - \frac{\mathbb{1}(l_i \neq z)p^*(l_{i+1} = y)}{1 - p^*(l_i = z)}\right), 0\right)$$

(3)

For the statement '$Z$ amplifies $Y$' for a sequence $k$, the loss is:

$$L_{tot} = -\sum_{i=1}^{n_k} p^*(l_i) + \alpha\, max\left(\frac{1}{n-1}\sum_{i=1}^{n-1}\left(\frac{\mathbb{1}(l_i \neq z)p^*(l_{i+1} = y)}{1 - p^*(l_i = z)} - \frac{\mathbb{1}(l_i = z)p^*(l_{i+1} = y)}{p^*(l_i = z)}\right), 0\right)$$

(4)

$\alpha$ provides a trade-off between two loss terms. Like many penalty constraint optimization approaches, when $\alpha \to \infty$, the optimal solution tends to satisfy the pairwise causal knowledge constraint; when $\alpha \to 0$, it is domain-knowledge-free. An advantage of such an incompatibility framework is that it is linearly additive, and therefore more statements can be easily incorporated as additional loss terms.

## 4.3 Outcome Model with Window $w > 1$

While the outcome model for the probability of occurrence of $y$ in the very next position ($w = 1$) can be easily computed for an autoregressive model, the task of computing this probability for $w > 1$ windows becomes combinatorial in nature. To illustrate this, note that the quantity $p(I(y)_i^w|h_{i-1}, l_i)$ or $1 - p(l_{i+1} \neq y, l_{i+2} \neq y, ..., l_{i+w} \neq y|h_i)$ can be computed (sequentially) as $1 - \prod_{k=1}^{w} p(l_{i+k} \neq y|h_{i+k-1})$ for $w = 1$. Some efficient estimation techniques are available from recent work for probabilistic querying [50] for larger $w$. Consider the selection of window $w$ for the outcome in an event sequence of length $n$. The problem in this case is relevant to the following probabilistic query – the probability of not observing any event $Y$ within the next $w$ events. Thus the (worst case) computational cost is $\mathcal{O}((|\mathcal{L}| - 1)^{n-1})$ for the largest window $w = n - 1$ in our setting, according to Definition 1. In this paper, we assume that $w$ is relative small compared to sequence length for the sake of practicality. ILE and SLE with respect to the statement '$Z$ reduces the occurrence of $Y$' for a sequence $k$ can be defined similarly for a window $w > 1$ and can be expressed as $p(I(y)_i^w|h_{i-1}, l_i = z) - p(I(y)_i^w|h_{i-1}, l_i \neq z)$ and $\frac{1}{n-w}\sum_{i=1}^{n-w}(p(I(y)_i^w|h_{i-1}, l_i = z) - p(I(y)_i^w|h_{i-1}, l_i \neq z))$, respectively.

Correspondingly, for a statement such as '$Z$ inhibits $Y$' and '$Z$ amplifies $Y$' for a sequence $k$, our proposed combined loss for $w = 2$ can be expressed respectively as

$$L_{tot} = -\sum_{i=1}^{n_k} p^*(l_i) + \alpha \, max(\frac{1}{n-2} \sum_{i=1}^{n-2} (\frac{\mathbb{1}(l_i = z)(1 - p^*(l_{i+1} \neq y)p^*(l_{i+2} \neq y))}{p^*(l_i = z)}$$
$$- \frac{\mathbb{1}(l_i \neq z)(1 - p^*(l_{i+1} \neq y)p^*(l_{i+2} \neq y))}{1 - p^*(l_i = z)}), 0) \quad (5)$$

$$L_{tot} = -\sum_{i=1}^{n_k} p^*(l_i) + \alpha \, max(\frac{1}{n-2} \sum_{i=1}^{n-2} (\frac{\mathbb{1}(l_i \neq z)(1 - p^*(l_{i+1} \neq y)p^*(l_{i+2} \neq y))}{1 - p^*(l_i = z)}$$
$$- \frac{\mathbb{1}(l_i = z)(1 - p^*(l_{i+1} \neq y)p^*(l_{i+2} \neq y))}{p^*(l_i = z)}), 0) \quad (6)$$

We perform some ablation experiments on the choice of $w$ (i.e. $w = 2$) in (sub)section 5.4.

## 5 Experiments

We implement our approach using a transformer architecture [51] and test the performance of our model on synthetic datasets as well as real-world datasets on the task of event prediction. We also explore a unique application where event concepts are generated from a large language model. Further details around implementation and training can be found in the Appendix.

### 5.1 Synthetic Experiments

**Datasets.** We run experiments on 4 generated synthetic datasets to verify the learning capabilities and validity of our approach. We simulate event sequence data using a binary summary Markov model (BSuMM), which assumes that the probability of an event label's occurrence at any position in a sequence depends on whether or not labels from its influencing set have occurred in some recent look-back period [23]. Here we consider simple event sequences involving BSuMM dynamics over 5 event labels, denoted A, B, C, D and E. For each experiment, we generated 5 samples of datasets where each dataset was split into train/dev/test sets $(60/20/20)\%$. We first used BSuMM to simulate 50 event sequences over the 5 event labels, each of length 100, where only two interactions occur: $B$ excites $A$ and $B$ inhibits itself. We incorporated the ground truth causal pair as $(B, A)$, with $B \nearrow A$. This makes up experiments for Synth-1 and Synth-2 where the look-backs (windows) used for generation are 2 and 4 respectively. Similarly, we generated another 2 sets of 50 sequences over the 5 event labels, each of length 100, with interactions as shown in Figure 3; these are the Synth-3 and Synth-4 datasets. Here we only consider the causal pair $(C, B)$ with ground truth $C \nearrow B$ as injected knowledge. In the Appendix, we show results for when other causal pairs are injected (one at a time) as knowledge.

**Baselines and Metric.** We consider 5 baselines to compare against our model. BSuMM and OS-uMM are two types of parametric summary Markov models that have been used for event prediction and influencing set identification [23]. Another common parametric model for event sequences is the k-th order Markov chain (kMC); for this baseline, we show the best performing version over $k \in \{1, 2, 3, 4\}$. We also consider a recent state-of-the-art transformer model – Probabilistic Attention to influence (PAIN) – that uses attention-to-influence techniques for modeling temporal event sequences [34]. Lastly, we include a vanilla transformer for event sequences (TES) [51] as a direct comparison to our proposed PC-TES with $\alpha \in \{100, 10, 1, 0.1, 0.01, 0.001\}$. We evaluate model performance using the metric of loglikelihood (LL) on the test subset, which is equivalent to a logarithmic scoring rule for the task of predicting the next event label given the history [52]. Further details such as hyper-parameters chosen based on the validation set are discussed in the Appendix.

**Results.** Results of the synthetic experiments are shown in Table 1. Our model PC-TES outperforms others by incorporating one pairwise statement. It is worth noting that the neural models considered here predict better than non-neural models by a significant margin, even with relatively small datasets.

Table 1: Next event prediction loglikelihood on 4 synthetic datasets.

| Dataset | BSuMM | OSuMM | kMC | PAIN | TES | PC-TES |
|---------|-------|-------|-----|------|-----|--------|
| Synth-1 | -382.82(4.67) | -373.85(6.37) | -364.60(6.47) | -141.06(5.11) | -107.81(2.85) | **-107.19(3.01)** |
| Synth-2 | -371.13(6.36) | -370.66(5.12) | -350.37(7.45) | -137.94(8.42) | -114.01(2.31) | **-111.89(1.52)** |
| Synth-3 | -358.25(7.23) | -359.678(12.30) | -378.97(5.64) | -131.20(12.79) | -119.99(2.29) | **-113.58(3.81)** |
| Synth-4 | -363.36(6.93) | -361.82(5.45) | -371.75(6.35) | -134.09(13.39) | -113.95(3.05) | **-113.15(3.66)** |

## 5.2 Real Applications

**Datasets, Baselines and Metric.** We perform experiments on the following 4 real-world applications from various domains shown in Table 2, referring the reader to dataset details in the Appendix. **Diabetes** contains meals, exercise activity, insulin dosage, and changes in blood glucose measurements for a group of 65 diabetes patients [53]. We incorporated the single statement 'insulin intake decreases blood glucose' into our objective, which agreed with the assessments mentioned in [47]. **Defi** contains user trading events including borrow, repay, deposit, redeem, liquidation and swap from the Aave website, aave.com. The sole qualitative statement in this study is: 'borrowing increases the probability of liquidation' [33]. **Linkedin** contains user-level employment-related events for LinkedIn users. We included all 10 event types as self-exciting, according to prior work [54, 23], which provided 10 qualitative statements. **Beigebooks** consists of a series of economic topics derived from documents issued by the Federal Reserve Board regarding the economic state of affairs in the US. We incorporated 8 causal statements about economic topics, such as 'vehicle sale robust' increases 'dealer inventory increase'. All datasets were split into train-dev-test sets $(60/20/20)\%$ for fair evaluation, where hyper-parameters were chosen using train-dev sets and evaluation was performed on the test sets. We consider the same baselines and evaluation metric.

Table 2: Dataset summary: # of event labels $(M)$, # of sequences $(K)$ and # of events $(N)$.

| Dataset | $M$ | $K$ | $N$ |
|---------|-----|-----|-----|
| BeigeBooks | 15 | 260 | 2370 |
| Diabetes | 13 | 65 | 20210 |
| LinkedIn | 10 | 1000 | 2212 |
| Defi | 6 | 500 | 17258 |
| LLM-Generated Event Sequences | 50 | 243 | 1398 |

Table 3: Next event prediction loglikelihood on 4 real-world applications.

| Dataset | BSuMM | OSuMM | kMC | PAIN | TES | PC-TES |
|---------|-------|-------|-----|------|-----|--------|
| Diabetes | -658.61 | -577.76 | -656.19 | -466.36 | -373.27 | **-370.64** |
| Defi | -1172.49 | -1152.84 | -1155.60 | -129.49 | -126.56 | **-120.54** |
| Linkedin | -114.14 | -115.05 | -111.39 | -25.32 | -22.48 | **-21.71** |
| Beigebooks | -40.21 | -40.82 | -46.09 | -21.077 | -10.27 | **-10.04** |

**Results.** Results on the 4 real-world benchmarks are shown in Table 3. Overall, our model PC-TES outperforms others by incorporating pairwise statements. We highlight again that results of neural models are substantially better than their non-neural counterparts. The highest performance boost by PC-TES occurs on Defi where the relative gain is around $5\%$ compared to a vanilla transformer model TES, validating the practicality of our approach for incorporating domain knowledge.

## 5.3 LLM-Generated Event Sequences

**Motivation.** We explore a use case of event prediction where we have *partial* causal knowledge about the events, along with powerful generative large language models (LLMs) to generate event sequences. It is challenging to use LLMs to generate event sequences, since depending on the choice of parameters (e.g., *temperature* which controls the stochasticity of the generation process), one is able to either generate a large number of very noisy sequences or a small number of high-quality sequences. A noisy collection will typically result in much lower precision in a downstream application, while a small collection will result in a loss of recall. Here we explore a solution that leverages a small but

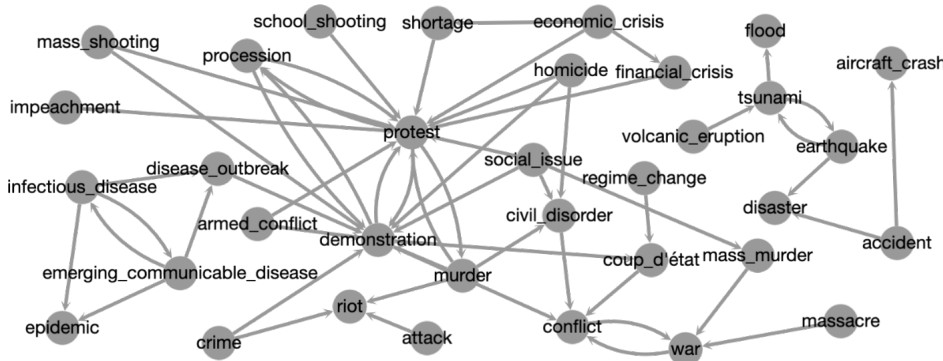

Figure 2: Causal knowledge graph of Wikidata event concepts.

```
[demonstration, regime_change, coup_d'état, looting, civil_disorder]
[impeachment, statutory_law, regime_change, coup_d'état, civil_disorder]
[earthquake, tsunami, disaster, looting, conflict]
[arson, explosion, mass_murder, crime]
[school_shooting, civil_disorder, riot, social_issue]
[attack, mass_murder, riot, civil_disorder, war, regime_change]
[energy_crisis, shortage, economic_crisis, civil_disorder, protest]
```

Figure 3: Examples of LLM-generated event concept sequences.

high-quality set of generated sequences along with an incomplete source of knowledge to produce a high-accuracy prediction model.

**Dataset.** For this application, we use an LLM (`flan-t5-xxl(11B)`) to create sequences of event concepts from Wikidata [55]. These are high-level concepts (classes) in Wikidata that represent newsworthy event types. We gather these types by querying Wikidata for concepts that had links to Wikinews articles and belonged to a subclass of the `occurrence` class, indicating that they are newsworthy event classes. We compile the common top-level classes of these concepts, which yielded 50 event-related Wikidata classes. Some of these have causal relations (e.g., `has_cause` or `has_effect`) either directly (e.g., earthquake causes tsunami) or through their instances (an earthquake event causing a tsunami event). Using these relations, we build a causal knowledge graph, depicted in Figure 2. The pairs in this network are used for knowledge injection.

To generate event sequences, we first create a prompt that restricted the output to the 50 event concept labels and asked for the next event concept in a sequence in the form of a question: "What usually happens after X?" for an input event concept X. For subsequent events, we remove the already-seen concepts from the vocabulary and use the generated event concept to ask the next question in a similar way. We repeat the questions until we reach a pre-defined maximum sequence length, or if the LLM fails to generate an in-vocabulary response in $k$ number of attempts. For the experiments in this paper, we use a random number between 2 and 10 as the maximum sequence length and $k = 10$ for the number of LLM generation attempts. We repeat this sequence generation procedure 5 times for each of our 50 event sequences, which resulted in 243 sequences (with 7 failed generation attempts). Figure 3 shows examples of the generated event sequences.

**Results.** Table 4 demonstrates how 3 neural models perform prediction on the LLM generated dataset, with and without 5 representative pairwise causal relations in different domains, from as natural disaster to social activities. We demonstrate that with our approach, without any pretraining, we can boost predictive performance by 3-4% as compared to a transformer event model. This shows the effectiveness of our approach in building event models with limited or no event sequences, using LLMs in conjunction with knowledge injection.

### 5.4 Ablation on Window for Outcome

We perform ablation experiments with window size in Table 5 ($w = 2$) on the synthetic simulations generated by the two different graphs from section 5.1 and Diabetes dataset. Overall, results from $w = 2$ do not differ significantly from $w = 1$, yet consistent improvements over TES are still observed.

Table 4: Next event prediction on LLM generated event sequences.

| Causal Pair | LL |
|---|---|
| None (PAIN) | -33.10 |
| None (TES) | -28.29 |
| impeachment ↗ protest (PC-TES) | -28.19 |
| tsunami ↗ earthquake (PC-TES) | -27.48 |
| mass_shooting ↗ protest (PC-TES) | -28.10 |
| infectious_disease ↗ epidemic (PC-TES) | -28.21 |
| murder ↗ riot (PC-TES) | -27.81 |
| All pairs above (PC-TES) | **-27.37** |

Table 5: Next event prediction on synthetic datasets with window $w = 2$.

| Dataset | LL |
|---|---|
| Synth-1 | -115.04(2.06) ↑ |
| Synth-2 | -113.27(2.50) ↓ |
| Synth-3 | -107.65(3.33) ↓ |
| Synth-4 | -111.38(2.78) ↑ |
| Diabetes | -370.60 ↑ |

## 6 Conclusion

In this paper, we addressed the under-explored area of incorporating pairwise causal relations into deep learning models for temporal event sequences. We proposed a novel approach that injects qualitative causal knowledge into transformer-based models and demonstrated a significant enhancement in performance. Our established framework for causal inference in temporal event sequences provides a theoretical justification that ensures unbiased estimation of the proposed measure. Through extensive experimentation, we validate the superior performance of our proposed approach over several state-of-the-art models on a probabilistic prediction task. By effectively leveraging knowledge about causal pairs, our approach showcases its ability to capture the intricate relationships between events and make enhanced predictions in multivariate event sequences. Furthermore, we introduce a unique application where we leverage a large language model to generate sequences of societal events. By demonstrating how a causal knowledge graph can be used for event prediction within such sequences, we highlight the practical implications of our framework. In summary, our work offers a promising direction for improving the performance of transformer-based models in multivariate event sequences by explicitly incorporating pairwise causal information.

### Broader Impact and Limitations

Our approach is centered around event sequences that may span a considerable length of time and involve potentially repeated instances of all event types. We operate under the assumption that the influence of past events are confounding, while ensuring robustness against confounding factors through strong ignorability. However, we advise caution when employing causal pairs, as incorrect causal knowledge may potential lead to incorrect conclusions. We highlight that the availability of knowledge that is consistent with the ground truth in general is crucial to performance in models that involve knowledge augmentation.

### Acknowledgments and Disclosure of Funding

This work was supported by the Rensselaer-IBM AI Research Collaboration (http://airc.rpi.edu), part of the IBM AI Horizons Network (http://ibm.biz/AIHorizons). We express our gratitude to the anonymous reviewers and meta-reviewer for their valuable feedback, which contributed to improving the quality of our work.

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
