# Appendix: Pairwise Causality Guided Transformers for Event Sequences

## 1 Synthetic Generators

We generated datasets from Binary Summary Markov Models (SuMM) where the generating dynamics is characterized by the instantiation of binary parental states for a specific look-back window [1]. We describe the parameters used in our experiments to generate four temporal event datasets namely synth-1, synth-2, synth-3, synth-4 over labels A,B,C,D and E in our paper. The graphs are shown in Figure 1. In summary, BSuMM-1 generates synth-1 and synth-2; and BSuMM-2 generates synth-3 and synth-4 respectively.

The parameters for BSuMM-1 are the following probabilities: $p_A = \{B = 0 : 0.2, B = 1 : 0.6\}$ $p_B = \{B = 0 : 0.6, B = 1 : 0.2, \}$ $p_C = 0.15$, $p_D = 0.025$, $p_E = 0.025$, respectively. In synth-1, we use a window of 2 and in synth-2, we use a window of $4$. The window determines the binary instantiation of the parental states for a particular event type. For example, in synth-1, for event $A$, if an event of $B$ is observed in the previous window of 2, the probability of observing $A$ for the current position is 0.6 otherwise it is 0.2. Noting the probabilities of all events sum up to 1, at each position, we generate an event from a categorical distribution. Similarly the parameters for BSuMM-2 are the following probabilities: $p_A = \{(B = 0, C = 0) : 0.6, (B = 0, C = 1) : 0.2, (B = 1, C = 0) : 0.7, (B = 1, C = 1) : 0.4\}$ $p_B = \{(B = 0, C = 0) : 0.2, (B = 0, C = 1) : 0.6, (B = 1, C = 0) : 0.1, (B = 1, C = 1) : 0.4\}$ $p_C = 0.15$, $p_D = 0.025$ and $p_E = 0.025$ respectively. In synth-3, we use a window of 2 and in synth-4, we use a window of $4$.

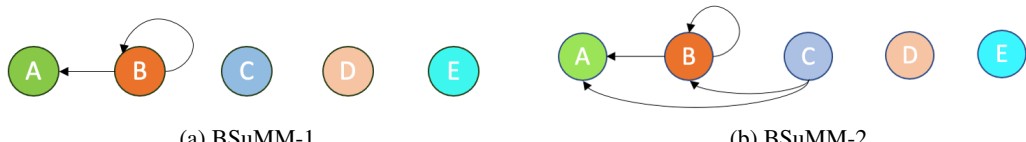

(a) BSuMM-1                    (b) BSuMM-2

Figure 1: Two BSuMM graphs are used to generated the four synthetic datasets: BSuMM-1 generates synth-1 and synth-2; and BSuMM-2 generates synth-3 and synth-4 respectively.

## 2 Real Application Dataset Details

The 5 real-world applications cover various domains. A descriptive summary of the 5 datasets used in our experiment are given in Table 1. Further details about the curation of Beigebooks, Diabetes and LinkedIn follow [1]. The Defi dataset provides user-level cryptocurrency trading history under a specific protocol called Aave. The original curated dataset from early work [2] includes timestamp, transaction type and coin type for each transaction. To ensure relevance and applicable to our temporal event sequence, we remove unnecessary features for our study, specifically timestamp and

Submitted to 37th Conference on Neural Information Processing Systems (NeurIPS 2023). Do not distribute.

coin type, and thus resulting in a dataset that contains only sequences of events of the following transaction types: borrow, repay, deposit, redeem, liquidation and swap.

| Dataset | $M$ | $K$ | $N$ |
|---|---|---|---|
| BeigeBooks | 15 | 260 | 2370 |
| Diabetes | 13 | 65 | 20210 |
| LinkedIn | 10 | 1000 | 2212 |
| Defi | 6 | 500 | 17258 |
| LLM-Generated Event Sequences | 50 | 243 | 1398 |

Table 1: Dataset summary: # of event labels ($M$), # of sequences ($K$) and # of events ($N$).

# 3 Model Implementation and Training

Our implementation of PC-TES model is based on the code adaptation from [3], and can be found in the Supplementary Material. We transformed the original temporal point process model, known as the Transformer Hawkes Process, into a vanilla Transformer for Event Sequence (TES) model by replacing the temporal encoding with position encoding. In addition, similar to natural language understanding tasks, we minimize the negative log-likelihood of event sequences (event token) for TES (details in Section 5). We incorporate loss terms associated with incompatibility to generalize to our PC-TES model.

To train our model, we utilize stochastic gradient descent and employ the Adam optimizer for optimization. The default transformer architecture used for training is specified as follows: the number of layers in the multi-headed self-attention module (n_layer), the dimension of the value vector after attention (d_model), the number of attention heads (n_head), the hidden layer size of the feedforward neural network (d_inner), the dimension of the value vector (d_v), the dimension of the key vector (d_k), and the dropout rate.

Experimental parameters for all datasets are provided in Table 2, which correspond to the results obtained during our experiments. It is important to note that the final parameters were selected based on the best performance of the model, determined by the minimum loss on the dev subset during evaluation. All experiments were conducted on a private server equipped with a TITAN RTX GPU.

Table 2: Hyperparameters for PC-TES for all datasets in the experiments. Synth represents all datasets generated by BSuMM, namely synth-1, synth-2, synth-3,synth-4

| Parameter Value | Synth | Beigebooks | Diabetes | LinkedIn | Defi | LLM-generated event sequences |
|---|---|---|---|---|---|---|
| batch_size | 32 | 32 | 16 | 32 | 16 | 32 |
| n_head | 4 | 4 | 6 | 4 | 6 | 4 |
| n_layers | 4 | 4 | 6 | 4 | 6 | 4 |
| d_model | 64 | 128 | 128 | 64 | 128 | 256 |
| d_inner | 128 | 256 | 256 | 128 | 256 | 512 |
| d_v | 64 | 256 | 128 | 64 | 128 | 256 |
| d_k | 64 | 256 | 128 | 64 | 128 | 256 |
| dropout | 0.1 | 0.1 | 0.1 | 0.1 | 0.1 | 0.1 |
| epoch | 500 | 500 | 500 | 500 | 500 | 500 |
| learning_rate | 0.0006 | 0.0002 | 0.0006 | 0.0004 | 0.0006 | 0.0001 |
| $\alpha$ | 10 | 0.001 | 1 | 0.001 | 0.1 | 0.001 |

# 4 Baseline Model Implementation Details

We provide details about the implementation of the baseline models.

**k-th order Markov chain (kMC).** We implement a simple $k^{th}$ order Markov chain over $k = \{1, 2, 3, 4\}$. Prior work has shown that prediction performance deteriorates on these event sequence datasets beyond $k = 4$ [1]. Only the results for the best performing $k$ is shown in the tables.

To ensure that the model learns probabilities that are not $0$ while evaluating the test set, we take a Bayesian approach to parameter learning using a Dirichlet prior with a single hyper-parameter $\alpha_D$. We use the following hyper-parameter grid to choose the optimal hyper-parameter using the train/dev sets: $\alpha_D \in \{0.1, 1, 5, 10\}$.

**Summary Markov models (BSuMM and OSuMM).** We learn binary and ordinal summary Markov models using the score-based approach in [1]. We use the following hyper-parameter grids to choose optimal hyper-parameters using the train/dev sets: Dirichlet hyper-parameter $\alpha_D \in \{0.1, 1, 5, 10\}$, look-back $\kappa \in \{1, 3, 5, 10\}$, complexity penalty $\gamma \in \{0.1, 0.5, 1\}$.

**Transformer for Event Sequences (TES).** We implement with Pytorch a $B$-block attention-based transformer network to model the dynamics and seek to maximize the log-likelihood of event sequences $\mathbf{D}$ in Equation 1:

$$log p_\theta(\mathbf{D}) = \sum_{k=1}^{K} \sum_{i=1}^{N_k} log p_\theta^*(l_i) \tag{1}$$

We use an un-modified history representation $\mathbf{H}^{(B)}$ from the $B^{th}$ block and the generated labels can be modeled via a multinomial distribution:

$$\theta_\psi(l_{i+1} = m | \mathbf{H}^{(B)}(i)) = \frac{\exp(\mathbf{W}_{m,:}\mathbf{H}^{(B)}(i) + \mathbf{b}_m)}{\sum_{m=1}^{M} \exp(\mathbf{W}_{m,:}\mathbf{H}^{(B)}(i) + \mathbf{b}_m)} \tag{2}$$

where $\mathbf{W}_{m,:}$ is the $m^{th}$ row of the corresponding trainable weight matrix and $\mathbf{b}_m$ is $m^{th}$ entry of the corresponding bias term. We perform prediction experiments and hyperparameters are selected from the best performing model from dev set, similarly to PC-TES as shown in Table 2.

**Probabilistic Attention-to-Influence Neural Model (PAIN).** PAIN is an innovative model that has been recently introduced to analyze temporal event sequences [4]. One of its key strengths is its ability to capture intricate instance-wise interactions between events, while also uncovering the influencers for each event type of interest. To accomplish this, PAIN leverages event sequence data and a prior distribution on type-wise influence. Through the proposed approach, PAIN efficiently learn an approximate posterior for type-wise influence by employing an attention-to-influence transformation with variational inference. Furthermore, this method goes on to model the conditional likelihood of sequences by sampling from the derived posterior. This sampling strategy enables the model to selectively focus attention on the event types that have the most significant impact on the overall sequence. A minor modification is made to make a fair comparision in our study, the origin random **vector** $\mathcal{V}$ is changed to a binary random **matrix** $\mathcal{A}$ which describes the pairwise interactions between all pairs of events, and thus the modified variational loss is:

$$\mathbb{L}(\omega, \psi; \mathbf{D}) = \mathbb{E}_{q_\omega}[\log \frac{p(\mathcal{A})}{q_\omega(\mathcal{A}|\mathbf{D})}] + \mathbb{E}_{q_\omega}[\log \theta_\psi(\mathbf{D}|\mathcal{A})] \tag{3}$$

where $p(\mathcal{A})$ $q_\omega(\mathcal{A}|\mathbf{D})$ are the prior and posterior paramterized by the $\omega$ network, and $\theta_\psi(\mathbf{D}|\mathcal{A})$ is the loglikelihood term given a sampled matrix $\mathcal{A} \sim q_\omega(\mathcal{A}|\mathbf{D})$. The prior used in our experiment is $p(\mathcal{A}) = \prod_{XY} p(\mathcal{A}_{XY} = 1) = 0.2$ for any $X, Y \in \mathcal{L}$ where $\mathcal{L}$ is the label set. Training details are consistent with the setting described in the paper [4].

# 5 Synthetic Experiments with Other Causal Pairs

We provide additional results on synthetic experiments where we use a different set of causal pair statements. For synthetic experiments synth-1, synth-2, we consider the causal pair $(B, B)$ with ground truth $B \searrow B$ as injected knowledge. For synthetic experiments synth-3, synth-4, we consider 3 additional causal pairs $(B, B), (C, A), (B, A)$ with ground truth $B \searrow B$, $C \searrow A$ and $B \nearrow A$ as injected knowledge, respectively. Results in Table 3 show that the overall knowledge injection via our approached significantly boosts predictive performance in all cases. Yet different sets of causal pair statements may enhance differently partial due to the optimization trajectory which we will explore in the future.

Table 3: Next event prediction loglikelihood on 4 synthetic datasets. Italics indicates improvement over TES; bold indicates the best performance.

| Dataset | BSuMM | OSuMM | kMC | PAIN | TES | PC-TES ($C \nearrow B$) | ($B \nearrow A$) | ($C \searrow A$) | ($B \searrow B$) |
|---|---|---|---|---|---|---|---|---|---|
| Synth-1 | -382.82(4.67) | -373.85(6.37) | -364.60(6.47) | -141.06(5.11) | -107.81(2.85) | ***-107.19(3.01)*** | N/A | N/A | -112.04(3.60) |
| Synth-2 | -371.13(6.36) | -370.66(5.12) | -350.37(7.45) | -137.94(8.42) | -114.01(2.31) | *-111.89(1.52)* | N/A | N/A | ***-110.78(2.15)*** |
| Synth-3 | -358.25(7.23) | -359.678(12.30) | -378.97(5.64) | -131.20(12.79) | -119.99(2.29) | *-113.58(3.81)* | *-103.02(3.35)* | ***-102.87(3.45)*** | *-103.02(3.36)* |
| Synth-4 | -363.36(6.93) | -361.82(5.45) | -371.75(6.35) | -134.09(13.39) | -113.95(3.05) | *-113.15(3.66)* | *-105.52(3.62)* | ***-105.31(3.67)*** | *-105.51(3.63)* |

## 6 Causal Inference Assumptions

Our approach builds upon the well-established potential outcomes framework [5], and its extensions to incorporate time-varying treatments and outcomes [6]. This framework has been widely utilized in previous studies that share a similar objective to ours in observational longitudinal studies [6, 7, 8, 9]. To identify a counterfactual outcome distribution over time, or more precisely, the average $w$-step-ahead potential outcome conditioned on history as defined in Definition 1 in the main text, it is necessary to make three standard assumptions regarding the data generating mechanism. These assumptions are crucial for establishing causal inference and enable us to estimate the potential outcomes in the presence of time-varying treatments and outcomes. Without loss of generality we consider $w = 1$ for a sequence $k$ and a causal pair (z,y), namely the 1-step-ahead potential outcome conditioned on history. For ease of notation, let $Z_i$ be binary random variable - $Z_i$ is 1 if $l_i = z$ else is 0, $Y_i^*$ be the outcome defined in Definition 1, namely the probability of occurrence (at least once) of event $Y$ in the next window $w = 1$ at position/time $i$. Let $\mathcal{H}_i$ be a random trajectory that generates the history. Let $Y_{i+1}^*[Z_i = 1]$ be the potential outcome under binary treatment at $i$.

**Assumption 1 (Consistency).** When a specific unit (such as a user or patient) receives a given sequence of treatments denoted as $l_i = z$, it follows that the potential outcome $Y_{i+1}^*$ under the treatment sequence $l_i = z$ is equal to the observed outcome. In other words, the potential outcome aligns with the factual outcome for the patient when considering the specific condition $l_i = z$.

**Assumption 2 (Sequential Overlap).** Throughout the entire history space over time, there is always a non-zero probability of both receiving and not receiving any treatment: $0 < p(Z_i = 1|\mathcal{H}_{i-1} = h_{i-1}) < 1$ for all $i \in \{1, ..., N_k\}$ for some realization of history $h_{i-1}$.

**Assumption 3 (Sequential Ignorability).** Conditioned on the observed history, the current treatment is independent of the potential outcome: $Z_i \perp Y_{i+1}^*[Z_i]|\mathcal{H}_{i-1}$ for all $i \in \{1, ..., N_k\}$. This independence implies that there are no unobserved confounding factors that simultaneously influence both the occurrence of the outcome $Y_i$ and the treatment assignment $Z_i$.