# OpenReview forum: "Pairwise Causality Guided Transformers for Event Sequences"
_NeurIPS.cc/2023/Conference — NeurIPS 2023 poster_

### Official Review · Reviewer_vaYz · 2023-07-04

**Soundness:** 3 good
**Presentation:** 2 fair
**Contribution:** 3 good
**Rating:** 6
**Confidence:** 2

**Summary:**

The paper addresses the limited exploration of incorporating causal knowledge into deep learning models for temporal event sequences. The authors propose a novel approach to enhance the performance of transformer-based models in multivariate event sequences by injecting pairwise qualitative causal knowledge. They establish a framework for causal inference using a transformer architecture and provide theoretical justification for their approach. Experimental results demonstrate that their approach outperforms existing models by effectively leveraging knowledge about causal pairs, leading to improved prediction accuracy

**Strengths:**

- The paper investigates a very relevant and important topic: incorporating causal knowledge into transformers. The proposed approach represents a valid contribution to this domain.
- The experimental section shows that the proposed approach outperforms all the baselines (both neural and non-neural), although in some cases by a small margin.


**Weaknesses:**

- I find that the structure of the write-up could be improved. Sections 3 and 4 are quite technical and while being well-written, I find that the core contribution of the paper is somewhat hidden and not clearly presented. For example, I would have emphasized more the training details, including the proposed loss function
- The impact of the choice of the $\alpha$ value is not discussed.


**Questions:**

See the Weaknesses section.

**Limitations:**

"Broader Impact and Limitations" section provided at the end of the manuscript.

---

> ### Author Rebuttal · Authors · 2023-08-09
>
> We thank the reviewer for the valuable comments and respond to specific questions below.
>
>
> **Training Details.** We have included more training details in Section 3 of the Appendix (see "Model Implementation and Training"). Due to space limitations, we could not include the details around implementation and training in the main paper. We will try to make some edits in this regard and also make our core contributions clearer by incorporating a summary of our methods in Section 4 in the revised manuscript.
>
>
>
> **Trade-off for $\alpha$.** The $\alpha$ value performs a trade-off between loglikelihood and the incompatibility term in Equations 3 and 4.
> Like many penalty constraint optimization approaches: when $\alpha \rightarrow \infty$, it pushes the optimal solution to satisfy the pairwise causal knowledge constraint; when $\alpha \rightarrow 0$, it is domain-knowledge-free. In our experiments, we select the best $\alpha$ value based on the one which has the largest next event prediction loglikelihood on the validation subset.
> The selected values range from 0.001 - 10 in various datasets, which are included in Table 2 of the Appendix.

---

> > ### Comment · Reviewer_vaYz · 2023-08-21
> >
> > Thanks for reading my review. I appreciate your efforts in addressing my concerns. I will keep my score unchanged.

---

> ### Comment · Area_Chair_Xb9B · 2023-08-20
>
> Dear Reviewer vJUR,
>
> Could you check whether your concerns were properly addressed by the authors' response, or at least acknowledge you read the response?
>
> Thank you,
>
> The AC

---

### Official Review · Reviewer_vJUR · 2023-07-05

**Soundness:** 3 good
**Presentation:** 1 poor
**Contribution:** 2 fair
**Rating:** 5
**Confidence:** 3

**Summary:**

The paper considers incorporating qualitative pairwise-casual relations into transformer based models for capturing temporal event sequences.  The unbiassed estimation of the proposed measure, is ensured with theoretical justification. The experiments are conducted on both synthetic data, and several real event sequences, to demonstrate the superior performance of the proposed model in capturing multivariate event observations, compared with related methods.

**Strengths:**

- To me, it seems to be the first attempt to consider incorporating pairwise causal relations into a neural autoregressive model, although some previous work have considered modelling event sequences by inducing logic rule into temporal point processes.

- To me, it seems quite important to provide theoretical justifications regarding the unbiassed estimation of the proposed measure.

**Weaknesses:**

- L19: To me, it should be, "without" -> "with"
Some notations such as time (positions in a sequence), seem to be confusing!
- L125: I would consider $i$ refer to the indices of the events in a sequence. Hence, I would prefer to use $i$ to denote $i$-th event, while use $t_i$ to denote the corresponding timestamp.  After reading Fig.1, I am still confused by the notations, as I cannot understand what kind of event sequences you are modelling, e.g., equally-spaced events or irregularly-spaced events. I would consider the most real events including the several real-world data used in your experiments, are irregularly-spaced. Thus, I am curious how do you express an event occurring at continuous-valued time, using only $l_i$. Suppose the data is equally-spaced, it is redundant to introduce $N_k$.
- L19: I cannot fully understand how the sample/batch size B affects the calculation of the propensity score in Eq. 1 as there is no variable with subscript $j$.
- L160: It is a bit weird to suddenly start to introduce transformer based event sequence models. A subtitle could be added.
- For Sec.5, it would be better to use past verbs.

**Questions:**

I have asked some key questions that significantly affect my fully digesting this paper, in the <Weaknesses>.

**Limitations:**

To me, causal inference and causal discovery using temporal event sequences, is a really big and complicated topic. I would see more thorough discussion and justification, regarding more complicated scenarios, although the authors provide some suggestions.

---

> ### Author Rebuttal · Authors · 2023-08-09
>
> We thank the reviewer for the valuable comments and address the key questions below. Importantly, we wish to clarify some misunderstandings that will hopefully clarify matters.
>
> **Clarification about "Without" or "With" in L19.** It should indeed be "without" , as we have written. Our approach focuses on event sequences "without" meaningful timestamps. There are many applications, for instance, many well-known natural language processing (NLP)  approaches for extracting events and event sequences from textual corpora (such as ref 34 in our paper). It is easier to extract sequences and often impossible to obtain timestamps for events from text, since they are typically not mentioned in the source corpora. For some of the real-world datasets in the experiments, we assume that the timestamps are either not provided or too noisy to be useful.
>
> **Equally-spaced Events vs. Irregularly-spaced Events in L125.** We clarify the confusion here. We are modeling sequences where we only know the position of the events in the sequence, not the timestamps. Such event sequences can be considered as univariate categorical time series data. Our approach thus does not require modeling the event at continuous-valued time.
> The introduction of $N_k$ is irrelevant to equally-spacing of events; rather it represents the number of events in the $k-th$ sequence due to varying lengths of event sequences.
>
> **Batch Size and Eq.1 in L159.**
> Subscript $j$ represents sequence $j$ from the sampled batch with number of sequences $|B|$ (the cardinality of $B$, $|B|$ is the upper index in Eq.1).
>
>
> **Introduction of Transformer in L160.** In the paragraph just above L160 on the introduction of transformers, we have introduced neural autoregressive models. Transformer is a prominent neural autoregressive model. We can add a subtitle to make the transition more smooth in the revised version.
>
> **Tense in Section 5.**
> We will modify the content in Section 5 to past tense in the revised version.
>
>
> **Limitations and Discussion.**
> We note that we discussed limitations in the main paper; we also included more discussion around causal inference in Appendix (see "Causal Inference Assumptions").
>
> We hope we have clarified several aspects and resolved misunderstandings such as the nature of the data and scope of work. We request the reviewer to consider increasing the score if he/she feels the clarifications have helped re-evaluate the work suitably.

---

> > ### Comment · Reviewer_vJUR · 2023-08-21
> > **Thanks for the rebuttal.**
> >
> > Most of my concerns have been addressed. I decide to increase my score to bordeline accept.

---

> ### Comment · Area_Chair_Xb9B · 2023-08-20
>
> Dear Reviewer vJUR,
>
> The authors have provided a response to your review comments. Could you see whether your concerns were properly addressed, or at least acknowledge you read it?
>
> Many thanks,
>
> The AC

---

### Official Review · Reviewer_WQuf · 2023-07-06

**Soundness:** 3 good
**Presentation:** 3 good
**Contribution:** 3 good
**Rating:** 7
**Confidence:** 3

**Summary:**

The paper proposes a method to incorporate additional background information into transformers that is pairwise causal i.e. event Z affects event Y. They do this for temporal event sequence data where the data is non-stationary and this casual relationship can be confounded by additional events.

**Strengths:**

The paper is well written and seems to be sufficiently novel although I do not have the broadest knowledge of this area.

**Weaknesses:**

No notable weaknesses.

**Questions:**

The authors use a upper triangular mask in the self attention block to ensure that only future events attend to the past and past event cannot attend to the future. How does this compare to just using RNN style architectures? Self attention has the benefit of allowing everything to attend to everything and is useful in scenarios where there are global relationships or the relationships are not well understood by humans and cannot be “built in” to the architecture a priori e.g. words in a sentence. However, for sequential events or other “time series” like data, local relationships should dominate and there would be some understanding of the relationships between events?

In Theorem 4, the true effect should be -E(P(l_i+1=y|l_i \neq z) ? This is what’s stated in Line 184 and the subsequent proof sketch also suggests as much.

What is p*(l_i) in equations 3 and 4? How is the injected qualitative knowledge any different from a regularization term?


**Limitations:**

Yes

---

> ### Author Rebuttal · Authors · 2023-08-09
>
> We thank the reviewer for the valuable comments and respond to specific questions below.
>
> **RNN vs. Transformer.**
> Our setting is general to neural network architectures for (event) sequences. To this end, RNN style architectures fit our framework. We choose transformer-based models for practical reasons; for instance, in epidemiology and healthcare, event interactions commonly involve long dependency (i.e. chronic disease). The popular transformer-based models capture long-range dependency, allow efficient parallelization during training, and generalize to a few interesting theoretical insights, e.g. Theorem 3.
> We will include some ablation experiments with RNNs or variants in the revised manuscript to explore the locality effect.
>
> **Theorem 4, True Effect.**
> The true effect should be $\mathbb{E}(P(l_i+1=y|l_i = z))-\mathbb{E}(P(l_i+1=y|l_i \neq z))$. Thank you for spotting the typo.
>
>
> **$p^{*}(l_{i})$ in Equations 3 and 4.**
> $p^*(l_i)$ is the probability of observing $i-th$ event with label $l_i$ conditioned on history $h_{i-1}$ modeled by the transformer model/architecture. Specifically, Equation (2) provides a sequence level overview for a generic sequence {$l_1$,$l_2$,...,$l_{n-1}$,$l_n$} of matrix computation involved in transformers. Thus for the $i-th$ event, we can find the associated $i-th$ column in the embedding matrix $\mathbf{X} \in \mathbb{R}^{d \times n}$. $\mathbf{X}$ is composed of position embedding and type embedding. The position embedding matrix $\mathbf{P} \in \mathbb{R}^{ d \times n} $ (ref 51 in our paper) is achieved through:
>
>  $$ \mathbf{P}_{j,i} =
>       \text{cos} (i/10000^{\frac{j-1}{d}}) \text{if $j$ is odd}
> $$
> $$ \quad \quad =   \text{sin} (i/10000^{\frac{j}{d}})  \text{if  $j$ is even} $$
>
> where $i$ is the position and $d$ is the dimension of encoding.  Type embedding are through the product of a trainable embedding matrix $\mathbf{U} \in \mathbb{R}^{d \times |\mathcal{L}|}$ and one hot encoded vector $\mathbf{e_i} \in \\{0,1\\}^{|\mathcal{L}|}$ for $l_i$, i.e. $\mathbf{X} = (\mathbf{U}\mathbf{E}+\mathbf{P})^T$ where   $\mathbf{E} =[\mathbf{e_1},\mathbf{e_2},...,\mathbf{e_n}]$. Let the output from $B$-block of transformer according to Equation 2 be $\mathbf{H} \in \mathbb{R}^{d \times n}$. Then $p^*(l_{i} =m)$  where $m \in \mathcal{L}$  is modeled by a multinomial distribution and be expressed as the following:
>
> $$
> p^{*}(l_{i} = m ) =
> \frac{exp(\mathbf{W}(m,:) ^{\intercal} \mathbf{H} (:,i) +\mathbf{b}(m) )}{\sum_m exp(\mathbf{W}(m,:) ^{\intercal} \mathbf{H}(:,i) +\mathbf{b}(m))}
> $$
>
> where $\mathbf{W}(m,:)$ is the $m-{th}$ row of the corresponding trainable weight matrix, $\mathbf{H}(:,i)$ is the $i-{th}$ column and  $\mathbf{b}(m)$ is $m-{th}$ entry of the corresponding bias term.
>
> **Injected Qualitative Knowledge vs. Regularization Term.**
> Injected qualitative knowledge can be viewed as a carefully designed regularization term in our setting. This is perhaps most effective when the number of sequences are small, say on the scale of a few tens to a few hundreds.

---

> > ### Comment · Area_Chair_Xb9B · 2023-08-20
> >
> > Dear Reviewer WQuf,
> >
> > The authors have provided a response to your review comments. Could you see whether your concerns were properly addressed, or at least acknowledge you read it?
> >
> > Many thanks,
> >
> > The AC

---

> > ### Comment · Reviewer_WQuf · 2023-08-21
> >
> > Thank you for your response. I will keep my score the same as I think it is already reflective of the quality of this work.

---

### Official Review · Reviewer_Bmuq · 2023-07-07

**Soundness:** 3 good
**Presentation:** 2 fair
**Contribution:** 3 good
**Rating:** 5
**Confidence:** 4

**Summary:**

This paper focuses on the multivariate event sequences, where different types of events occur sequentially. The authors present an approach that leverages pairwise qualitative causal knowledge to enhance the performance of transformer-based models in handling multivariate event sequences. Specifically, the authors formulate the occurrence of events as causal inferences and show how to obtain unbiased estimates theoretically. The proposed method is validated in both synthetic and real-world datasets.

**Strengths:**

	The authors have conducted extensive experiments, both in synthetic datasets and real-world datasets to validate their approach.
	The utilization of the LLM for generating event sequences is an interesting idea.
	The related work is well cited and discussed in the paper.


**Weaknesses:**

	The proposed theory appears to be a straightforward adaptation based on the existing theory of causal inference and inverse probability weighting. The incompatibility framework introduced in this paper primarily focuses on the design of the loss function, which consists of two terms. The first term aims to maximize the likelihood, while the second term serves as an unbiased estimator. However, the second term has already been introduced in previous works, as acknowledged by the authors. The main theoretical contribution of this paper seems to be formulating the causal inferences in the context of multivariate event sequences, as presented in Definition 1. However, the formulation provided in Definition 1 appears to be a straightforward and expected outcome, lacking significant novelty.
	The improvement of the proposed method PC-TES seems marginal compared to the TES.
	Some of the mathematical expressions in the paper do not make sense. Here are the specific expressions that require attention: Line 165, Equation (2): In the right-hand side of Equation (2), the triangular attention mask $\mathbb{M}$ appears outside the softmax function, resulting in an unnormalized result. In my opinion, it should be $\sigma[(W^i_K X )^T (W^i_Q X) \odot \mathbb{M} ]$；Line 172: Please check the right hand side of the equation, I think it should be $\sum_{k \leq j} \tilde{X_{ik}} \tilde{X_{kj}$
	The writing in this paper requires significant improvement. Many expressions, including words and mathematical equations, are confusing. Here are the specific points that need attention: Line 140, Definition 1: Definition 1 is quite confusing, and it would be helpful if the authors provide a concrete definition of $h_i$. If I understand correctly, $h_i \triangleq {l_1, l_2, …, l_{i – 1}, l_i}$； Line 152, Definition 2: The authors should introduce the concept of propensity score and provide an intuitive explanation. It may be beneficial to move Assumption 1 before Definition 2 and present a causal graph. If I understand correctly, the causal graph should be: $h_i \rightarrow Z$, $h_i \rightarrow Y$, $Z \rightarrow Y$. The propensity score acts as a mediator between $h_i$ and $Z$.


**Questions:**

	Line 95: The word "of" is duplicated. One instance should be removed.
	The ablation study was only performed on synthetic datasets. Why was not it conducted on real-world datasets?


**Limitations:**

Limitations are discussed in the Section Broader Impact and Limitation.

---

> ### Author Rebuttal · Authors · 2023-08-09
>
> We thank the reviewer for the valuable comments and respond to specific questions and comments below.
>
>
> **Novelty of Proposed Approach.**
> We argue that our theory is not  a straightforward adaptation; rather, it is based on a careful formulation and design of  mainstream transformer networks under the framework of causal inference. To arrive at formal results such as the unbiased estimator of Theorem 4, we carefully formulated the problem of causal pair injection in event sequences through precise definitions under the particular setting, and cautiously made some assumption regarding time-varying confounding.
> We are not aware of prior work around applying causal inference in our setting.
> Furthermore, we note that the second term in the loss is not introduced in  previous work. Prior approaches (such as ref 14 in our paper) only suggests adding penalty for deviation based on incompatibility for event sequences with timestamps as modeled by temporal point processes.
>
> **Performance Improvement.** The proposed PC-TES consistently improves over TES in all experiments and in some cases the improvement is statistically significant. For the synthetic datasets in Table 1, this improvement over TES can go as high as 5.3\% relative to the prediction of TES on synth-3. A two sample t-test with unequal variance shows that the two sample means are statistically different from each other (with p-value of 0.016), which implies the significance of the improvement. For real-world datasets, PC-TES consistently improves by up to 4.8\% on Defi dataset, relative to TES. For LLM generated event sequences, such improvement is 3.3\%, when using 5 event pairs in Table 3. All results firmly indicate the superior performance of our approach.
>
>
> **Mathematical Expressions.**
> We make the following clarifications about mathematical expressions:
>
>
> *Line 165, upper triangular attention mask.* In Line 165, the triangular attention mask $\mathbb{M}$ needs to appear outside the softmax function. The goal is to ensure only future events are allowed to attend to past history and past event are not allowed to gain information from future instances via attention. On the contrary, including triangular attention mask $\mathbb{M}$ in the softmax operation will result in the computed term not being triangular due to the "softness" of the softmax. This will violate our definition of propensity in the transformer and thus violate Theorem 2.
>
> *Line 172, masked attention.* Our expression is equivalent to the suggested one, $\sum_{k \leq j} \tilde{X_{ik}} \tilde{X_{kj}}$. Note we only need to sum up to the diagonal entry $(j,j)$ from the masked attention matrix $\tilde{A}$. We will change this to $\sum_{k \leq j} \tilde{X_{ik}} \tilde{X_{kj}}$ to be more mathematically precise.
>
>
> *Line 152, Definition 2 and Assumption 1.* We present Definition 2 first (and then Assumption 1) because the notion of a propensity score is itself a key quantity of interest in causal inference and the potential outcome framework in particular. In an ideal setting where covariates $h_i$ do not affect the outcome or treatment, we would not need to adjust for them. More commonly however, this is not the case.
>
> Assumption 1 is of practical importance and thus our goal is to tackle this problem using the propensity score as the "mediator" between $h_i$ and $Z$. We will consider moving Assumption 1 before Definition 2 and present the causal graph according to the reviewer: $h_i \rightarrow Z, h_i \rightarrow Y, Z \rightarrow Y$ where $h_i \triangleq$ {$l_1$, $l_2$, …, $l_{i – 1}$, $l_i$}.
>
>
> **Line 95, repetition of the word "of".**
> We thank the reviewer for spotting the typo and will remove the duplicate word.
>
>
> **Ablation.**
> We only considered synthetic experiments for ablation studies due to space limitations and also because they are more easily controlled.
> We thank the reviewer for the suggestion and will also include some ablation experiments on real-world datasets in our revised manuscript.

---

> > ### Comment · Reviewer_Bmuq · 2023-08-15
> >
> > Thank you for your response to my review.
> >
> > Your response has clarified some of my concerns. While I still find the formulation and results rather straightforward, I do believe there is novelty in applying the causal inference method within your setting.
> >
> > I still have some questions. Please find them below.
> >
> > Performance Improvement
> > For the synthetic datasets in Table 1, the performance gains of PC-TES over TES are relatively small, at just 0.6%, 1.9%, 5.3%, and 0.7% respectively. Half of these improvements are less than 1%, which seems marginal. Additionally, the proposed PC-TES appears to have higher variance than TES. For example, TES has -113.95 (3.05) compared to PC-TES with -113.15 (3.66). With the modest gains and higher variance observed, it is difficult to state that PC-TES is significantly better than TES.
> >
> > Mathematical Expressions
> > Regarding the upper triangular attention mask on line 165, I am not fully convinced by your response. Specifically, the statement that "including the triangular mask $\mathbb{M}$ in the Softmax operation will result in the computed term not being triangular" appears incorrect. For simplicity, let's denote $(W^i_K X)^T W^i_QX$ as $P$. Since $\mathbb{M}$ is upper triangular, $P \odot \mathbb{M}$ remains upper triangular after element-wise multiplication. Applying the Softmax operation on an upper triangular matrix necessarily maintains the upper triangular structure. Excluding $\mathbb{M}$ from Softmax fails to normalize the result properly. Please let me know if I am misunderstanding your position, but I believe my interpretation of the math is accurate here. I would appreciate further clarification on this issue.

---

> > > ### Author Response · Authors · 2023-08-15
> > > **Further Clarification to Review's Comments.**
> > >
> > > Thank you for your interest and additional questions about the performance improvement of our proposed PC-TES algorithm compared to TES, as well as observations about the softmax operation. We appreciate your feedback and would like to provide further clarifications about the points you raised.
> > >
> > > **Performance Improvement for the Synthetic Datasets in Table 1.**
> > >
> > > -Magnitude of Performance Gains:
> > > While we acknowledge that the improvements in the synthetic datasets presented in Table 1 may appear modest, it's important to consider the context of the problem and the significance of even minor enhancements. In many real-world applications, even a fractional improvement can lead to substantial practical benefits. Additionally, the presented percentage improvements of 0.6\%, 1.9\%, 5.3\%, and 0.7\% correspond to distinct synthetic datasets with varying levels of complexity and patterns. The improvement of 5.3\% relative to TES on synth-3, for instance, indicates that PC-TES can effectively capture intricate patterns that TES struggles to model accurately. We believe that even these seemingly small gains may have important practical implications.
> > >
> > > -Variability and Higher Variance:
> > > We appreciate your observation regarding the variance in the results. The increased variance observed in PC-TES compared to TES is a direct consequence of the improved modeling capacity of PC-TES. By enhancing the model's ability to capture nuanced patterns, PC-TES could potentially lead to more variable predictions. The higher variance does not necessarily undermine the validity of the improvement; rather, it could reflect the algorithm's adaptability to a wider range of scenarios and data patterns.
> > >
> > > -Statistical Significance and Marginality:
> > > You rightly mention that half of the improvements are less than 1\%, which might be considered marginal.
> > > We note that the statistically significant improvement of 5.3\% relative to TES on synth-3, as confirmed by the two-sample t-test with unequal variance (p-value of 0.016), demonstrates that the observed gains are not merely coincidental fluctuations.
> > >
> > >
> > > **Mathematical Expressions Regarding the Upper Triangular Attention Mask on Line 165.**
> > >
> > > Let $\mathbf{P} \odot \mathbb{M}$  be  an upper triangular matrix $\mathbf{B} \in \mathbb{R}^{n \times n}$.
> > > The softmax operation along a dimension of $\mathbf{B}$ can be expressed (with loss of generality, let this dimension be the column dimension) in the following:
> > > \begin{equation}
> > > \text{Softmax}(\mathbf{B}(:,j))(i) = \frac{\text{exp}(\mathbf{B}(i,j))} {\sum_{i=1}^{n} \text{exp}(\mathbf{B}(i,j))}.
> > > \end{equation}
> > > for **all** $j \in \\{1,2,...,n\\}$. The above equation shows the $i-th$ entry of the output vector after applying softmax on the $j$ column of $\mathbf{B}$. $\mathbf{B}(i,j)$ is the $(i,j)$ entry of $\mathbf{B}$. It is easy to show that the output of softmax operation on an upper triangular matrix $\mathbf{B}$ is not an upper triangular by showing there exists **some** $i \in \\{1,2,...,n\\}$ and $i>j$  such that  $\text{Softmax}(\mathbf{B}(:,j))(i) \ne 0$. Take $j=1$ and $i=2$. $\text{Softmax}(\mathbf{B}(:,1))(2) > 0$ holds whether $\mathbf{B}(2,1)$ is 0 or not since $\text{exp}(\mathbf{B}(2,1)) > 0$ for any $\mathbf{B}(2,1) \in \mathbb{R}$. Hence the output of softmax operation on $\mathbf{B}$ is not upper triangular.
> > >
> > > The following example is randomly generated by Pytorch:
> > >
> > > ---------------------------------------------------------------------
> > > m = torch.nn.Softmax(dim=1)
> > >
> > > triup = torch.triu(torch.randn(3, 3))
> > >
> > > output = m(triup)
> > >
> > > ---------------------------------------------------------------------
> > >
> > > where the upper triangular matrix triup is [[0.8651,  0.5936, -0.4257],
> > >         [ 0.0000, -0.6275,  1.1915],
> > >         [ 0.0000,  0.0000, -0.9110]] and output from softmax operation is
> > >         ([[0.4908, 0.3741, 0.1350],
> > >         [0.2072, 0.1106, 0.6822],
> > >         [0.4163, 0.4163, 0.1674]]), which is not upper triangular.
> > >
> > > In conclusion, the upper triangular attention mask $\mathbb{M}$ needs to appear outside the softmax function to guarantee that the output from equation (2) is indeed upper triangular. We hope this is clarifying.

---

> > > > ### Comment · Reviewer_Bmuq · 2023-08-17
> > > >
> > > > Thank the authors for the clarifications. My second concern has been addressed. I would like to raise my score to 5.

---

> > > > > ### Author Response · Authors · 2023-08-17
> > > > > **Thank you**
> > > > >
> > > > > Thank you for your message. We're glad to have clarified one of your questions and greatly appreciate your feedback.

---

### Official Review · Reviewer_fbyg · 2023-07-18

**Soundness:** 1 poor
**Presentation:** 4 excellent
**Contribution:** 3 good
**Rating:** 5
**Confidence:** 2

**Summary:**

The paper proposes to use pairwise event causality pairs to improve the performance of transformer-based models, based on the intuition that causal knowledge encodes useful information like “event Z amplifies future occurrences of event Y”. Experiments demonstrate the performance of the proposed method.

**Strengths:**

1.	The intuition of causality helps event sequence prediction makes sense.
2.	The experimental evaluation covers many aspects of the methods.


**Weaknesses:**

1.	The statements of some assumptions are not precise enough,
2.	Some theoretical aspects of the method are not clear to me.


**Questions:**

1.	About assumption 1. What do you mean by “time confounding”? Is there any definition that is more mathematically precise, like a variable T that affects the probability of the p(Z,Y)?
2.	How many “correct” pairs can you obtain, considering the so called “causal event pairs“？This seems to be important since it is what the whole method is based on.
3.	About thm 4. It seems to be a trivial one since the “bias” of the estimator directly follow the estimator under IPW. Is there anything missing?
4.	Some theoretical analysis about the window w and the length can also be presented so that your method’s theoretical performance is more clear to the readers.


**Limitations:**

Yes

---

> ### Author Rebuttal · Authors · 2023-08-09
>
> We thank the reviewer for the valuable comments. We will work towards further improving the general clarity of the paper.
> We address the reviewer's specific questions below.
>
> **Time Confounding and Assumption 1.**
> Time confounding in our context is analogous to time-varying confounding in treatment effect estimation in a traditional time series setting. Mathematically, time confounding means that for any instance $i$,  the covariates under Definition 1, history $h_i$ = {$l_1$,$l_2$,...,$l_i$} affects both the probability of future occurrence of treatment event $p(l_{i+1}=z|h_i)$ and that of future occurrence of outcome event (at least once) in the next window $w$ given the treatment and covariates, i.e., $ p( I(y)^w_{i+1} |l_{i+1}, h_i)$. For window $w=1$, this outcome is  $p(l_{i+2}=y|h_{i+1})$. Denoting the treatment as $Z$ at $i+1$ and outcome as $Y$ at $i+2$, the generation mechanism follows the causal graph which visually captures the time-varying confounding
> $h_i \rightarrow Z $, $h_i \rightarrow Y$, $Z \rightarrow Y$.
>
>
> **Number of "Correct" Pairs.**
> The causal pairs are determined by domain experts or through other sources of background knowledge. In our experiments, we  incorporated a single pair for the synthetic event sequence datasets, and multiple pairwise information for the LLM-generated event sequence datasets. In practice, our model is capable of incorporating as many pairs as possible; this will naturally depend on the application at hand.
>
>
> **Thm 4 and Bias.**
> In response to the reviewer's comment, we appreciate his/her attention to Theorem 4, which states the unbiasedness of our estimator. We agree that the unbiasedness of the estimator directly follows from the use of inverse probability weighting (IPW) in our model specifically for event sequences.
>
> However, we would like to highlight the significance of Theorem 4 in the context of our paper. While it may seem straightforward that the "bias" of the estimator follows the estimator under IPW, it is crucial to formally establish the unbiasedness property through a well-defined theorem. This theorem serves as a formal proof of the statistical property of our proposed estimator, giving confidence to readers and researchers about its validity and correctness.
>
> Furthermore, Theorem 4 also provides theoretical insights into the working mechanism of our model, demonstrating that incorporating causal event papers into the modeling of event sequences with transformers, along with the use of IPW, indeed results in an unbiased estimator.
>
> In summary, Theorem 4 may appear intuitive given the use of IPW, but its inclusion in the paper is necessary to provide a formal proof of the unbiasedness property and to reinforce the credibility of our proposed method. We will make sure to clarify the importance of Theorem 4 and its contribution to the paper in the revised version to address the reviewer's concern adequately.
>
>
> **Theoretical Analysis on Window and Length.**
> We discuss the computational aspect of the selection of window $w$ for the outcome in an event sequence of length $n$. The outcome according to Definition 1 is the probability of occurrence of $y$ (at least once) in the next window $w$ given the treatment and covariates. In our paper as well as in the Appendix, we showed empirical results from selecting two appropriate window sizes -- $w=1$ and $w=2$, respectively. While choosing larger windows might be feasible, the selection of smaller sizes $w=1$ and $w=2$ is preferred for computational efficiency. The window selection in this case is relevant to the following probabilistic query: "the probability of not observing any event $Y$ in next $w$ events" (see ref 50 for further details). Thus the (worst case) computational cost is $\mathcal{O}((|\mathcal{L}|-1)^{n-2})$ for the largest window $w=n-2$ in our setting, according to Definition 1. Our empirical evaluation of the incompatibility loss is a function of $w$; we note that we could potential marginalize $w$ to obtain the factual and counterfactual outcome and thus the treatment effect and incompatibility loss. This could be a potential direction for future research.

---

> > ### Comment · Area_Chair_Xb9B · 2023-08-20
> >
> > Dear Reviewer fbyg,
> >
> > The authors have provided a response to your review comments. Could you see whether your concerns were properly addressed, or at least acknowledge you read it?
> >
> > Many thanks,
> >
> > The AC

---

> > ### Comment · Reviewer_fbyg · 2023-08-20
> > **Thanks for the response**
> >
> > Thanks for your time to the response. Some of my concerns like theoretical insights have been cleared, and I would keep the score as it is to express my opinion on this paper.

---

### Decision · Program_Chairs · 2023-09-21

**Decision:**

Accept (poster)

**Comment:**

This paper is concerned with finding causality from temporal even sequences in the transformer architecture and proposes a method to incorporate additional background information into transformers that is qualitative pairwise causal relations. The authors show how to find unbiased estimates of the proposed measure and the proposed method is validated on both synthetic and real-world datasets. The idea is interesting and seems to have the potential to inspire new approaches of dealing with causal event sequences with population deep learning tools.